# Road User Attitudes and Their Reported Behaviours in Abuja, Nigeria

Uchenna Nnabuihe Uhegbu * and Miles R. Tight

Department of Civil Engineering, School of Engineering, University of Birmingham, Edgbaston, Birmingham B15 2TT, UK; m.r.tight@bham.ac.uk
* Correspondence: uchennauhegbu@gmail.com; Tel.: +44-7882592517

**Abstract:** The continuous increases in the numbers of road traffic crashes (RTC) over the years, especially in developing countries, have been a source of worry. The majority of the RTC are attributed to road user behaviours exhibited by the drivers. This study sets out to investigate the road user attitudes and behaviours in Abuja, Nigeria. A total of 1526 questionnaires were distributed, of which 321 questionnaires were completed and returned. The questionnaires tried to understand four major road user behaviours, namely use of seatbelts, drink driving, use of mobile phone while driving, and use of child restraints. The results after analysing the questionnaires showed that the majority of the road users in Abuja showed high non-compliance with the use of seatbelts, either when driving or when being driven. About 64% of the respondents admitted to not using seatbelts. Results from the cross-tabulation process showed that the high non-compliance to seatbelt usage was statistically associated with young, single road users. Around one-third of Abuja road users admitted to drink driving. Road users who were married engaged in more frequent drink driving than road users who were single, and the association was statistically significant. A high percentage of Abuja road users admitted to using mobile phones while driving and 55.8% of the total respondents admitted to not using child restraints while driving. A lack of child restraints was statistically associated more with male road users than female road users. It is recommended that stricter enforcement of road safety laws should be undertaken and that the government should provide road safety agents with the right equipment (e.g., speed guns, breathalyzers) that would aid road safety agents to perform their duties effectively in order to curb the excessive bad road user behaviours in Abuja.

**Keywords:** road traffic crashes; road user behaviours; road safety; crash fatalities; Abuja; Nigeria

## 1. Introduction

From global estimates, about 1.35 million lives are lost annually to injuries from road traffic crashes (RTC), leaving a global average of 20–50 million people with non-fatal injuries resulting in temporal or permanent disabilities [1]. The 2018 global status report on road safety also projects road traffic injuries to rank seventh in the global causes of death by 2030 [2].

Africa has its fair share of this global road safety crisis, and accounts for about 16% of global road fatalities, despite having only about 2% of the world's vehicles [3,4]. In Nigeria, the lead road safety agency, the Federal Road Safety Corps (FRSC), indicates that from 2011–2016, 69,941 crashes were recorded on Nigerian roads, resulting in 35,179 fatalities [5]. This is equivalent to a fatality every two road traffic crashes (RTC), or a daily average of 16 road traffic fatalities for the period under consideration (2011–2016). Of the 5053 fatalities recorded in 2016 in Nigeria, 93% (4696) were adults, while 7% (357) were children. Regarding gender, 79% (3970) were male, while (21%) 1083 were female [5].

In view of the overwhelming challenges generally affecting transportation in Nigeria, road safety research has only recently started gaining the desired attention from the

government. The Federal Government of Nigeria (FGN) is committed to the reduction of road traffic crashes (RTC), complimented by the delivery of a safe road transport system for all classes of road users. The government plans on achieving this through the Federal Road Safety Corps (FRSC) and the Federal Ministry of Transportation playing strategic and pivotal roles in the development of road safety policies. However, despite the laudable efforts of the FGN, road traffic crashes (RTC) and fatalities are still relatively high, which is a pointer to the fact that the country is yet to get it right [6,7].

A contributory factor to increased crashes in Nigeria is the increase in private vehicle ownership. The National Bureau of Statistics (NBS) [8] estimates the number of officially registered vehicles in Nigeria at 11,760,871 vehicles, of which 6,785,956 (57.7%) are commercial vehicles, 4,819,251 (40.9%) are privately owned, 149,470 (1.3 %) are government owned, while 6194 (0.1 %) are diplomatic vehicles. More importantly, 90% of road traffic crashes (RTC) are caused by road user behaviour [9,10]. Hingson et al. [10] also suggests that changes in driver behaviours offer the largest opportunity towards the reduction of road traffic crashes.

This study, therefore, analyses road user behaviours, with the aim of understanding the attitudes of road users toward road safety issues in Abuja, Nigeria. Prior to this work and within the literature identified, the authors were unable to find any previous studies focused on road user attitudes and behaviours responsible for the relatively high crash and injury rates in Nigeria's capital city of Abuja when compared to other Nigerian cities.

Section 1.1 looks at crash fatalities in Abuja, while the causes of road traffic crashes in Nigeria is discussed in Section 1.2. Section 2 broadly discusses the state of the art of this study. The adopted methodology is presented in Section 3, followed by the results and discussion in Section 4 and conclusion in Section 5.

### 1.1. Crash Fatalities in Abuja

In Nigeria, official road crash data were solely collected by the Nigerian Police Force (NPF) from 1960–1988 [11]; however, since the formation of the Federal Road Safety Commission (now Federal Road Safety Corps) in 1988, the Federal Road Safety Corps (FRSC) has been responsible for the collection of crash data nationwide. The annual FRSC crash data for all the 36 states in Nigeria and the Federal Capital Territory (Abuja) are aggregated, but are divided on a state-by-state basis into the numbers of fatal cases, serious cases, minor cases, total cases, persons killed, and persons injured, as well as total casualty. Mbakwe et al. [12] are of the opinion that Nigeria lacks a "reliable and comprehensive database" of traffic crashes and associated casualties. It is also known that road crash data are not readily available, and in very rare cases where they are available, they are underreported, incomplete, and lack the necessary information needed to tackle existing road safety problems [4]. This is consistent with the opinions of Iyanda [13] and Adeloye et al. [14]. Effective data collection; the management of crashes, injuries, and fatalities; as well as categorising vehicles involved in crashes are very important factors in understanding the traffic situation and proffering solutions to traffic safety challenges. Casado-Sanz et al. [15], in their investigation of the risk factors associated with the severity of injured pedestrians on Spanish crosstown roads, undertook a comparison of the crash databases of different countries. Table 1 shows a comparison of the Nigerian dataset with the earlier comparison by Casado-Sanz et al. [15], highlighting the weaknesses of the FRSC crash data.

Figure 1 shows the trend for crash fatalities in Abuja from 1990 to 2016. For the period under consideration, it can be observed that the annual crash fatalities are characterized by spikes and dips. The highest number of road fatalities recorded was 465 in 2011, while the lowest number of deaths recorded was 5 in 2001, which deviates largely from other recorded yearly road traffic fatalities in the city. There is no explanation given by Nigeria's lead road safety agency (the FRSC) or any available research that gives reasons for the sharp dips in fatalities experienced in 2001 and 2007, which were followed by rapid spikes in 2002 and 2008. This, therefore, typifies the data quality and reliability issues associated with the system of road traffic data collection in low- and middle-income countries (LMIC), including Nigeria.

**Table 1.** Comparison between official road crash databases.

| Variable | EU Directive [a] | US MMUCC [b] | Australia | New Zealand | Spain | Nigeria [f] |
|---|---|---|---|---|---|---|
| Crash location | Location as precise as possible | Road name, GPS coordinates | Road name, reference point, distance, direction | Road name, GPS coordinates | Road name, km | Route name |
| Crash narrative | No | No | Yes | Yes | Yes | No |
| Crash sketch | No | No | Yes, access restricted | Yes | Yes | No |
| Crash type | Yes | Recorded in the trafficunits section | Yes | Yes | Yes | Yes |
| Collision type | Yes | 8 descriptors | Yes | Yes | 33 descriptors | Yes |
| Contributing circumstances | No | Environmental circumstances | Yes | Yes | Yes | Yes |
| Weather conditions | Yes | 10 descriptors | Yes | 5 descriptors | 9 descriptors | Yes |
| Light conditions | Yes | 7 descriptors | Yes | 7 descriptors | Yes | Yes |
| Reported crashes | Not specified | All severities | All injury severities | All severities | All severities | All Severities |
| Definition of non-fatal injury levels | Severe and non-severe injuries | A: Suspected serious injury B: Suspected minor injury C: Possible injury | Injured, admitted to hospital Injured, required medical treatment | Serious: Requiring medical treatment Minor: other injuries | Hospitalised, injured Non-hospitalised, injured | Serious: Requiring medical treatment Minor: other injuries |
| Fatalities | Within 30 days | Within 30 days | Within 30 days | Within 30 days | Within 30 days | Within 30 days |
| Link with hospital data | No | No | In Western Australia | No | Yes | No |
| Contributing circumstances | No | 11 descriptors | No | Numerous cause codes | Yes | No |
| Speed limit | Yes | Yes | Yes | Yes | Yes [d] | Yes |
| Surface conditions | Yes | 10 descriptors | Yes | 3 descriptors | 9 descriptors | Yes |
| Road curve | No | Yes | Yes | 4 descriptors | 5 descriptors | No |
| Road segment gradient | No | Yes | No | No | No | No |
| Age | Yes | Date of Birth | Yes | Yes [c] | Yes | Yes |
| Gender | Yes | Yes | Yes | Yes | Yes | Yes |
| Nationality | Yes | No | Foreign drivers identified | Foreign drivers identified | Yes | No |
| Injury status | No | 5 descriptors | 4 descriptors | Yes | 5 descriptors | 2 descriptors |
| Driver action | No | 19 descriptors | In-crash narrative | In-crash narrative | 23 descriptors | No |
| Pedestrian action | No | 11 descriptors | In-crash narrative | In-crash narrative | 11 descriptors | No |

**Table 1.** *Cont.*

| Variable | EU Directive [a] | US MMUCC [b] | Australia | New Zealand | Spain | Nigeria [f] |
|---|---|---|---|---|---|---|
| Violation codes | No | Yes | Yes | Yes | No | Yes |
| Alcohol level | Yes | Yes | Yes | Yes | Yes | No |
| Drug test results | No | Yes | Yes | Yes | Yes | No |
| Safety equipment | Yes | Yes | Yes | Yes | Yes | No |
| Seating position | No | Yes | Yes | Yes | Yes | No |
| ADT [e] | No | Yes | No | Yes | No | No |
| Curve radius | No | Yes | No | Yes | No | No |
| Length | No | Yes | No | Yes | No | No |

[a] Directive 2008/96/EC of the European Parliament and of the Council of 19 November 2008 on Road Infrastructure Safety Management, [b] MMUCC Guideline. Model minimum uniform crash criteria. National Highway Traffic Safety Administration (NHTSA), [c] only pedestrian and cyclist ages in coded crash listing. Other ages in police crash reports, [d] speed limits recorded since 2015, [e] average daily traffic, [f] National Road Traffic Crash Data Management System (NRTCDMS). Source: Modified from Casado-Sanz et al. [15].

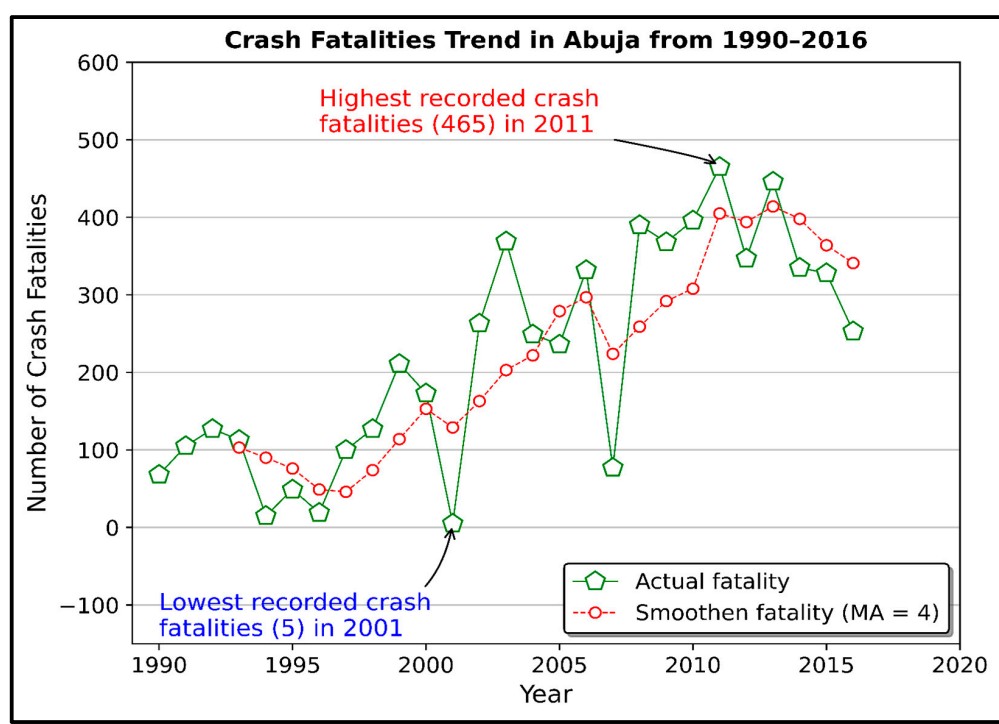

**Figure 1.** Crash fatalities trend in Abuja, from 1990–2016 [5].

Crash fatalities per 100,000 population is a good indicator of the severity of road traffic crashes, which takes into account the number of crash fatalities with respect to population. Figure 2 shows the average crash fatalities per 100,000 population in Nigeria from 2006 to 2016. It can be observed that compared to other states, Abuja had the highest rate of crash fatalities per 100,000 population. The fatality risk from road traffic crashes in Abuja was 20.54 per 100,000 population. This shows high levels of unsafety, despite the data quality issues and possible high levels of underreporting of crash data by the FRSC [2,16].

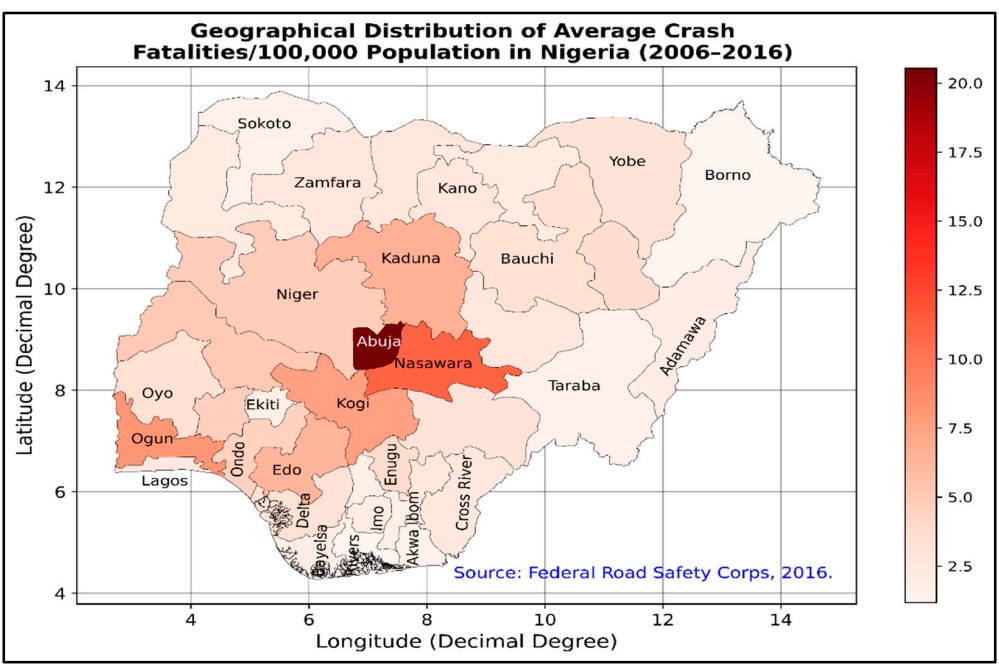

**Figure 2.** Crash fatalities per 100,000 population in Nigeria for 2006–2016 [5].

*1.2. Causes of Road Traffic Crashes in Nigeria*

Table 2 shows the main causes of road traffic crashes (RTC) in Nigeria according to the FRSC in 2016. The top 3 identified causes of crashes are speed-related factors (speed violation, loss of control, and dangerous driving), which accounts for 57.8% of all causes of crashes in Nigeria in 2016. The top 5 factors identified in Table 2 all relate to driving behaviour, which is a confirmation of the suggestion by Hingson et al. [10] that changes in driver behaviours offers the largest opportunity towards the reduction of road traffic crashes.

**Table 2.** Causes of road traffic crashes (RTC) in Nigeria.

| Causes | Number of Crashes | Percentage |
|---|---|---|
| Speed Violation | 3848 | 33.9% |
| Loss of Control | 1753 | 15.4% |
| Dangerous Driving | 969 | 8.5% |
| Wrongful Overtaking | 832 | 7.3% |
| Sign/Light Violation | 736 | 6.5% |
| Tyre Burst | 689 | 6.1% |
| Route Violation | 591 | 5.2% |
| Brake Failure | 567 | 5.0% |
| Mechanically Deficient Vehicle | 316 | 2.8% |
| Others | 246 | 2.2% |
| Road Obstruction Violation | 182 | 1.6% |
| Dangerous Overtaking | 144 | 1.3% |
| Bad Road | 124 | 1.1% |
| Overloading | 99 | 0.9% |
| Sleeping on Steering | 78 | 0.7% |
| Fatigue | 73 | 0.6% |
| Driving Under Alcohol/Drug Influence | 57 | 0.5% |
| Use of Phone While Driving | 32 | 0.3% |
| Poor Weather | 27 | 0.2% |
| **Total** | 11,363 | 100.0% |

Source: [5].

Several researchers have tried to understand the causes of road traffic crashes (RTC) in Nigerian cities. Uzondu et al. [4], in an observational study, reported that the incorrect use of indicators and tailgating were the two road user behaviours that were prevalent among road users in Owerri, Imo State, Nigeria. Sobngwi-Tambekou et al. [17], in a study on police reports in Cameroon, stated that mechanical failure, overtaking, and excessive speed accounted for road traffic crashes in Yaoundé–Doula, Cameroon. Afolabi and Kolawole [18], in a review of crash reports, stated that human, mechanical, and environmental characteristics are the salient factors of RTC in Nigeria. Findings from previous studies indicate that human factors rank high among the causes of road traffic crashes (RTC).

## 2. State of the Art

Several researchers have adopted various methods to try and understand road user behaviours, such as self-reported, naturalistic driving, and driving simulation methods, amongst others. Golias and Karlaftis, in a self-reported study of driver behaviour in 19 European countries, examined self-reported driver behaviours, including speeding, reckless

driving, seatbelt use, and drink driving [19]. Golias and Karlaftis reported that the compliance with the use of seatbelts increases with age and education for both men and women. Golias and Karlaftis also reported that Northern European drivers recorded a significantly higher compliance rate for seatbelt usage than Southern and Eastern European drivers.

Ortiz et al. reported that in 2015, driver distraction contributed to 3447 deaths and 391,000 injuries in the United States [20]. Ortiz et al., in an observational study of road intersections in the United States, stated that driver distraction could be caused by single or multiple distractors. Distractors such as engaging with other road users and the use of cell phones were the two behaviours exhibited the most by road users [20].

Stanojević et al. [21] adopted a driver behaviour questionnaire (DBQ) to study road user behaviours in three countries (Bulgaria, Romania, and Serbia) in Southeast Europe, and found that irrespective of cultural and socioeconomic differences, over speeding was exhibited the most by drivers in the three countries. Prat et al. [22], in an observational study of driving distractions on urban roads in Girona, Spain, observed 6578 drivers at nine randomly selected urban locations, and reported that the three most common distractions drivers experience are talking to a passenger, smoking, and mobile phone use.

Helman and Reed [23], in their study "Validation of the Driver Behaviour Questionnaire Using Behavioural Data from an Instrumented Vehicle and High-Fidelity Driving Simulator" reported that driving simulation is relatively valid, however the use of the DBQ is a valid measure of driving behaviour. Li et al. [24], in a driving-simulator-based study, reported that the use of cell phones by drivers tends to reduce driver brake reaction times. Xiaomeng et al. also reported that the use of mobile phones in hands-free mode does not eliminate the safety problems associated with distracted driving, as it impairs driving performance in the same way as hand-held mobile phones. Rumschlag et al. [25], in a driving simulator study, also reported that texting impairs driving performance, and that the effect is felt more in older drivers.

Several researchers have suggested that self-reported studies are not ideal in measuring road user behaviour due to the subjective nature and over-reporting of behaviours that would be considered socially unacceptable [23,26]. Naturalistic studies, which require that the road user behaviours be directly observed on the road, are considered true representations of driver behaviours; however, self-reported methods are still used due to the high costs associated with observational studies.

## 3. Materials and Methods

### 3.1. Study Area

Nigeria is Africa's largest economy [27], with an estimated gross domestic product (GDP) of $404.65 billion in 2016, and a gross national income (GNI) per capita of $2470, earning it a middle-income country status [28]. The choice of Abuja for this study is a no brainer, as it has the highest average number of crash fatalities per 100,000 population from 2006 to 2016, as shown in Figure 2. Abuja is Nigeria's capital city, situated in the middle of Nigeria, as shown in Figure 3, and covers an area of 7315 square kilometers. Between 2000 and 2010, Abuja grew by 139.7%, making it the fastest growing African city [29]. The population of the Federal Capital Territory (FCT)–Abuja was estimated to be 3,564,126 as of 2016 [30].

The Federal Capital Territory (FCT)–Abuja is bordered by four states, namely Nassarawa, Niger, Kaduna, and Kogi State (see Figure 3). The National Population Commission (NPC) estimates the 2016 population of Abuja's Municipal Area Council (AMAC) to be about 1.9 million. According to the National Bureau of Statistics, in 2016, Abuja had 950,202 registered vehicles.

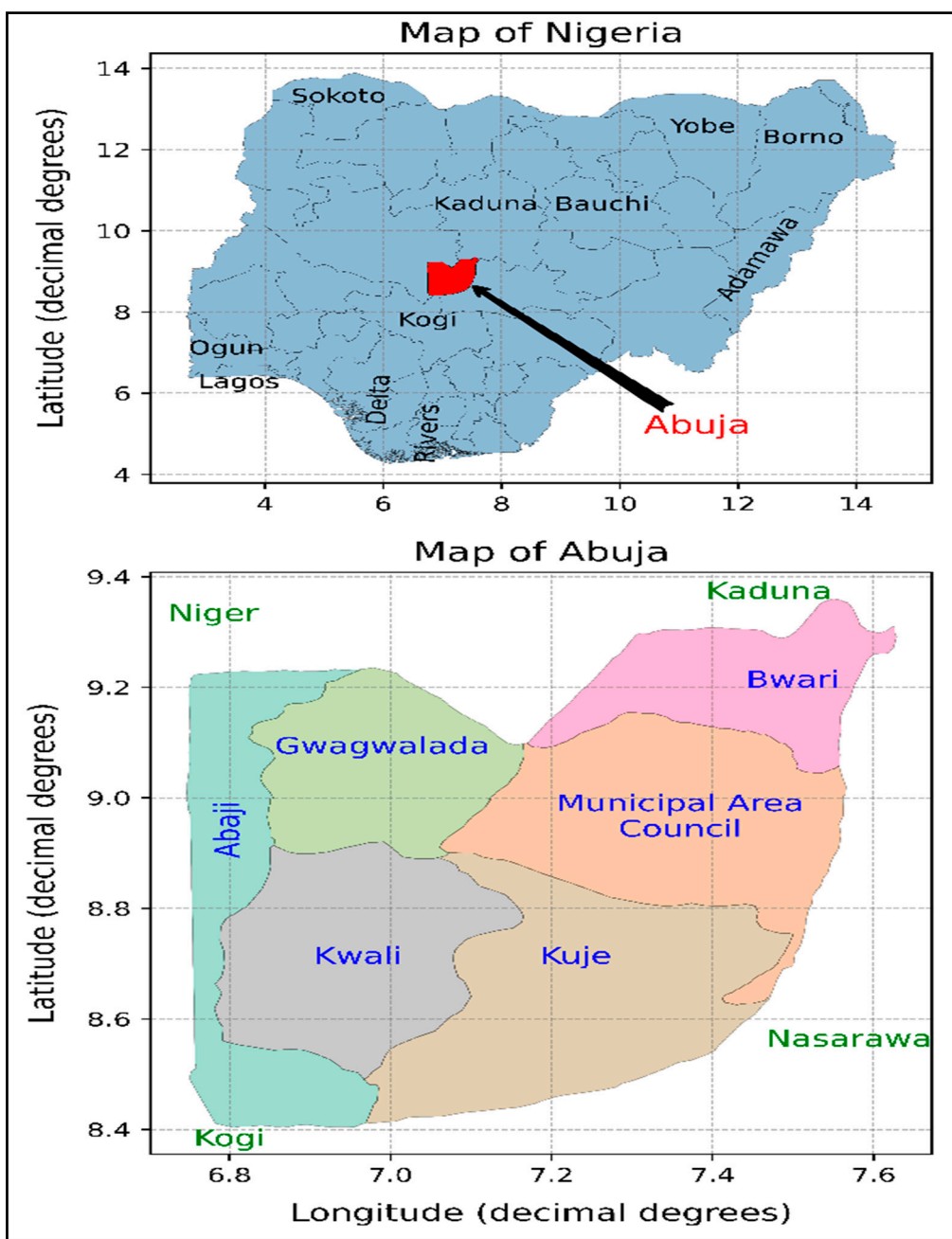

**Figure 3.** Map of Nigeria showing the location of Abuja.

### 3.2. Research Ethics

The adopted methodology was reviewed and granted full approval by the University of Birmingham's Science, Technology, Engineering, and Mathematics (STEM) ethical review committee. Permission to carry out the research was also granted by the relevant Nigerian authorities in Abuja.

### 3.3. Data Collection and Analysis

On request by the first author, Nigeria's Road Traffic Crash data for 1990–2016 were provided by the Federal Road Safety Corps (FRSC).

According to Adejugbagbe et al. [31], human behaviour and incapacitation is a common factor in about 85% of road traffic crashes (RTC) in Africa. It was decided to use a questionnaire to obtain information from road users (both drivers and pedestrians) re-

garding road user behaviours and attitudes. The questionnaires were produced in English language due to the fact that English language is the official lingua franca of the country.

Adopting a 95% confidence level and 5% margin of error, a sample size calculator was used in estimating the survey sample size for the road users in Abuja. In total, 1526 questionnaires were distributed face-to-face (by the first author) to road users, during a study period in Abuja from November 2018 to February 2019, and 321 (21%) (212 male and 109 female subjects) were successfully completed and returned. The questionnaire approach was adopted over an observational study due to the costs associated with observational studies and also due to how wide the roads are in Abuja, which makes the manual observation of drivers and other road users quite impossible from a distance.

The respondents are representative of both the male and female genders. Abuja is home mainly to the political and business classes and their retinue of aides and security operatives, which are mostly married males and above the age of 40. The federal civil service (the largest single employer of Abuja residents) is not exempt, as it is also male-dominated, requiring a minimum of secondary education to be employed. These demographics are also reflected in the number of females elected or appointed into both political and civil service offices in Abuja. Therefore, data obtained from the survey are representative of the population of Abuja. Data obtained are described in terms of gender, age, marital status, and highest education received. The survey campaign is complemented by data coming from the FRSC crash database (see Figures 1 and 2, Tables 1 and 2), which also serves as data source for this study and a subsequent study.

The questionnaires were administered with participant information leaflets and consent forms with the liberty to withdraw from the study at any time, even without reasons.

During the survey collection, the services of two assistants (male and female) were utilised strictly for the purpose of data collection (to distribute, administer, and collect fully completed questionnaires). Both assistants were necessary, as some respondents were more comfortable interacting with a specific gender. Both assistants had a good command of English language and Nigerian Pidgin English, and had the requisite interpersonal skills to easily interact with respondents. The assistants also made it a duty to clarify and respond to any concerns raised by the respondents.

The administered questionnaire was divided into four sections, namely personal data, road traffic crashes, driver behaviour, and road safety. In section A, personal data were used to obtain demographic information on the road users. Section B, road traffic crashes, tried to understand the frequency and nature of the road traffic crashes (RTC) that occur in Abuja. Section C, driver behaviour, tried to understand the road user attitudes and behaviours, while section D, road safety, tried to get a general sense of road safety in the city of Abuja. Sections B and C were vital to the study and were included in the study questionnaire because both give an understanding of the magnitude of road traffic crashes and what the contributing factors are in terms of road user behaviours that might be responsible for the high number of RTC in the city. Section D was also included in the study questionnaire to assess the performance of Nigeria's lead road safety agency (FRSC) with respect to generally improving road safety.

The successfully completed and returned questionnaires were analysed using univariate analysis to summarise and describe the responses, followed by a bivariate analysis to understand the relationship between categorical variables. Inferential statistics was also used in passing judgement on various road user behaviours the questionnaire aimed at measuring. The statistical software SPSS was used in carrying out the analysis.

## 4. Results and Discussion

### 4.1. Demographic Characteristics of the Sample

The demographic features of the respondents are presented in Figure 4. In total, 66% (212) of the 321 respondents were male, while 34% (109) of the respondents were female. Two sample tests of proportion showed that the male road users in Abuja were significantly greater than the female road users.

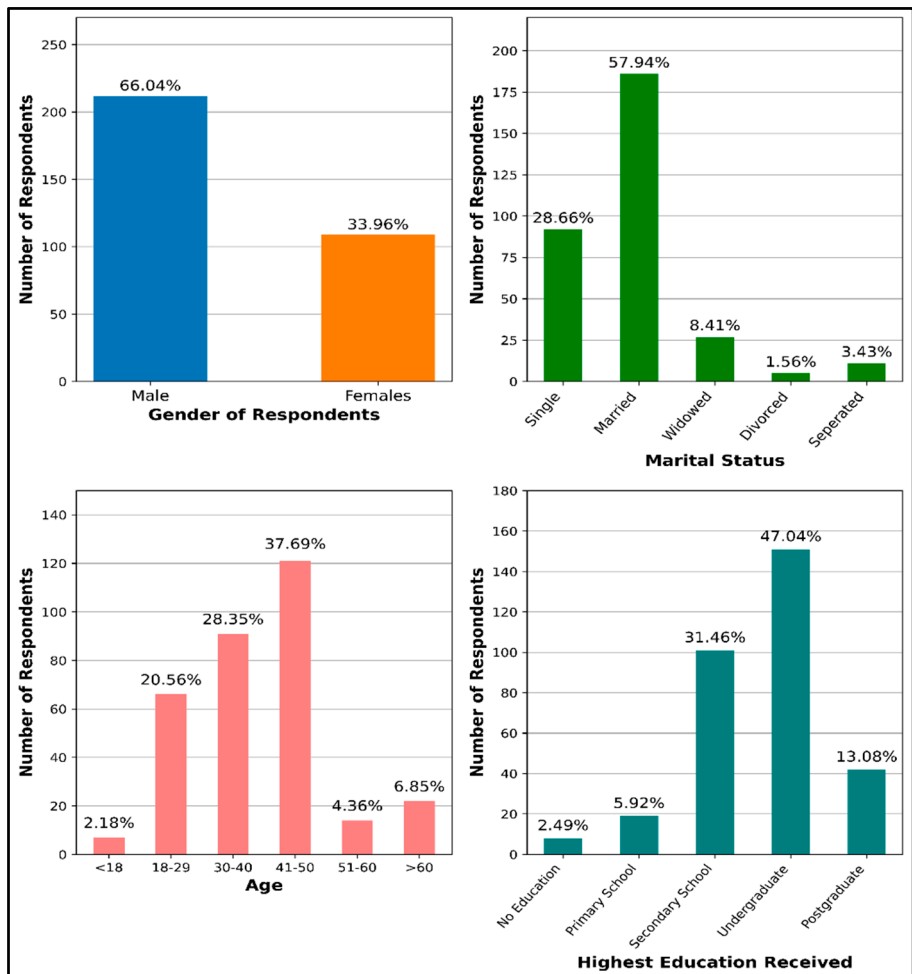

**Figure 4.** Demographics of the respondents.

Kumar and Srinivasan [32], in a study of the sociodemographic profile of road traffic crash victims in Karimnagar community in India, reported that more males drive automobiles than females and have a propensity to experience more road traffic conflicts than females. The data from the current study indicate that the majority of the respondents were married and more than half of the respondents were below 40 years, which was a good representation of the country's demographic. The majority of the respondents had received a minimum of secondary school education. With the high number of young road users in Abuja, there is an increased risk of road traffic crashes occurring. Aipoh [33] reported that about 65% of road traffic crashes are directly linked to youths between the ages of 15 and 45 years. Isa and Siyan [34] reported that more than 50% of all global road traffic fatalities occur among young adults between the ages of 15 and 40.

*4.2. Road Safety Concerns and Crashes in Abuja*

This study analyses road user attitudes and behaviours against the demographic criteria before establishing any relationship or making any inference about certain road user behaviours.

From the results of the study, 91% of male and 74% of female respondents reported that they were very bothered about the road safety situation and the occurrence of road traffic crashes in Abuja.

In total, 16.5% of male and 8.3% of female respondents admitted to having been involved in traffic crashes resulting in injury in the last 6 months. In contrast, 32.5% of the male respondents and 23.9% of the female respondents reported they had been involved in road traffic crashes in the last 6 months without any injuries. At a level of significance of

0.05, there were statistically significant relationships between involvement in a road traffic crash that resulted in injuries and gender ($p$ = 0.042), age ($p$ = 0.000), highest education received ($p$ = 0.011), and years of driving ($p$ = 0.002). However, there was no statistically significant relationship between marital status and involvement in road traffic crashes (RTC) that resulted in injuries in the last 6 months. The findings from this survey confirm findings reported from other studies that male drivers tend to be involved in more road traffic crashes than female drivers. Isa and Siyan [34] reported that 73% of fatalities from road traffic crashes (RTC) are associated with the male gender in the Kano–Kaduna–Abuja Dual Carriageway in Nigeria.

In total, 60.4% of male and 37.6% of female respondents indicated that in Abuja, most road traffic crashes (RTC) happen at night. This was statistically significant by gender ($p$ = 0.000), marital status ($p$ = 0.000), and years of driving ($p$ = 0.037). With respect to age, although the difference was statistically significant ($p$ = 0.000), the assumption for a minimum expected value of 5 was violated, therefore no relationship was shown.

### 4.3. Road User Behaviours

The key behavioural risk factors observed from the survey include those outlined below.

#### 4.3.1. Use of Seatbelts

About 64% of the respondents admitted to not using seatbelts when driving or when being driven in a car. The overestimation of seatbelt usage among road users in self-reported studies ranges from a factor of 1.2 to 2 when compared to observational studies [26,35,36]. The high non-compliance rate of seatbelt usage is not only common among road users in Abuja, but common in other Nigerian states and even amongst road users in other African countries. Several observational studies in Nigeria have shown that the non-compliance rates of seatbelt usage among road users in Ibadan, Benin city, Enugu, and Makurdi were as high as 81.3%, 47.7%, 62% and 72.7%, respectively [37–40]. Afukaar et al. [41] reported that the compliance rate of seatbelt usage in developing countries is less than 55%, which is a contrast from what is observed among road user in developed countries. According to Glassbrenner [42], the compliance rate for the use of seatbelts in the USA reached 75% in 2002, and continues to show a steady pattern of increase.

Table 3 shows that there were significant associations between the use of seatbelts and marital status ($\chi^2$ = 17.994, $p$ value < 0.001), age ($\chi^2$ = 22.48, $p$ value < 0.001), education received ($\chi^2$ = 21.23, $p$ value < 0.001), and years of driving ($\chi^2$ = 27.17, $p$ value < 0.001). Single respondents had lower seatbelt usage than married respondents. About 72% of single road users in Abuja admitted to the fact that they do not wear their seatbelts. About 71% of the respondents between the ages of 18 and 40 admitted to the fact that they do not wear their seatbelts. These observations are also similar in other studies, which observed higher compliance rates among elderly occupants [37,41,43,44]. The high non-compliance rates of seatbelt usage among single and younger road users might be due to youthful exuberance and downplaying the risks of injuries or even death from road traffic crashes.

The relationship between the use of seatbelts and highest education received showed that there was a direct relationship between the variables. Respondents with university degrees tend to wear their seatbelts more than respondents who just received primary education. The bivariate relationship also showed that drivers with fewer years of driving experience tend not to use their seat belt while driving. In total, 69.9% of the respondents with 0–10 years of driving experience admitted to not using their seatbelt while driving, while all the respondents with over 40 years of driving experience admitted using their seatbelt while driving. The high compliance rate of the use of seatbelt among drivers with more years of driving experience could be attributed to the fact that they must have developed the habit of wearing their seatbelts over the years. The high compliance rate of the use of seatbelts among experienced drivers could be attributed to the fact that they have either personally experienced or witnessed the advantages associated with wearing a seatbelt. High compliance rates of seatbelt usage while driving could be achieved through

proper legislation, enforcement, and publicity about the dangers associated with the non-use of seatbelts. However, enforcement of the use of seatbelts has been an issue in most developing countries, as the bodies responsible for carrying out that task are not able to do a proper job in that aspect. Agu et al. [39] attributed low compliance in seatbelt usage in Enugu, Nigeria, to the lack of enforcement of the seatbelt laws by the Federal Road Safety Corps (FRSC). This finding is also evident from the survey in Abuja.

**Table 3.** Chi-square results for road user behaviours for different demographic criteria.

| Demographic | Parameters | Road User Behaviours | | | |
| --- | --- | --- | --- | --- | --- |
| | | **Use of Seatbelt** | **Drink Driving** | **Use of Phone while Driving** | **Use of Child Restraint** |
| Gender | $\chi^2$ | 2.139 | 2.125 | 0.218 | 19.257 |
| | df | 1 | 2 | 1 | 1 |
| | *p*-values | 0.144 | 0.346 | 0.642 | 0.000 |
| Marital Status | $\chi^2$ | 17.994 | 19.507 | 2.669 | 0.664 |
| | df | 4 | 6 | 4 | 2 |
| | *p*-values | 0.001 | 0.003 | 0.615 | 0.718 |
| Age | $\chi^2$ | 22.48 | 7.343 | 5.218 | 2.650 |
| | df | 5 | 8 | 4 | 2 |
| | *p*-values | 0.000 | 0.500 | 0.266 | 0.266 |
| Highest Education Received | $\chi^2$ | 21.23 | 8.318 | 6.696 | 3.014 |
| | df | 4 | 8 | 4 | 3 |
| | *p*-values | 0.000 | 0.403 | 0.153 | 0.389 |
| Years of Driving Experience | $\chi^2$ | 27.17 | 11.221 | 6.674 | 5.777 |
| | df | 4 | 8 | 4 | 2 |
| | *p*-values | 0.000 | 0.189 | 0.153 | 0.056 |

### 4.3.2. Drink Driving

In total, 37.4% of the driving respondents admitted to have engaged in drink driving in the previous month when the survey was conducted. It was observed that married respondents engaged more frequently in drink driving compared to the single respondents. The percentage of married drivers that engaged in drink driving more than twice in the last month was more than the percentage of single drivers. The difference was statistically significant at the 0.05 level of significance ($\chi^2$ = 19.507, df = 6, *p* = 0.003). Other demographic criteria were not statistically significant regarding whether respondents engage in drink driving. The 2017 FRSC annual report [45] indicated that only 54 in 10,972 road traffic crashes (0.49%) were caused by "driving under the influence of alcohol or drugs". The very low number of crashes due to drink driving reported by the FRSC can be attributed to a lack of testing. Virtually no or little alcohol limit testing is randomly done on drivers or even when drivers are involved in crashes. Horwood and Fergusson [46] correlated drink driving to other general bad driving behaviours such as reckless driving and speeding. Therefore, it is possible for the FRSC to attribute RTC due to drink driving to other road user behaviours such as speed violation and dangerous driving. Judging by the percentage of drivers who admitted to having engaged in drink driving in the previous month when the survey was conducted in Abuja, there should be more alcohol testing on drivers in Abuja.

### 4.3.3. Use of Phone While Driving

Mobile phone use while driving can adversely affect road safety [47]. About 71% of the driving respondents admitted using their mobile phones while driving, which is illegal and contrary to the road safety laws of the country [48]. Similar findings were observed in other studies. Bener et al. [49], in a cross-sectional survey study, reported that 73.2% of Qatari drivers involved in a crash made use of their mobile phones while driving. Similarly, Olubiyi et al. [50], in a cross-sectional survey, reported that 86.6% of the motorists in Zaria, Nigeria, admitted to the use of their mobile phones while driving. Hill et al. [51], in a self-reported study of mobile phone usage while driving, reported that 69% of drivers in the Ukraine admitted to reading text messages while driving, while 49% admitted

to texting using mobile phone applications while driving. Findings from self-reported studies showed contradictory results to observational studies. For example, Ipingbemi and Oyemami [52], in an observational study, reported that the average mobile phone usage rate (while driving) among drivers in Ibadan, Nigeria, was just 4.2%. Alghnam et al. and Mahfoud et al. [53,54], in observational studies of the use of mobile phones while driving in Riyadh, Saudi Arabia, and Doha, Qatar, respectively, reported that 13.8% and 7.5% of drivers were observed using their mobile phones while driving. Findings on the use of mobile phone while driving in cross-sectional survey studies were higher compared to the findings from observational studies. This could be attributed to the fact that in observational studies, the behaviour of drivers can only be manually observed when they cross the observer's observation point, in contrast to when drivers voluntarily respond to questionnaire surveys about their mobile phone usage while driving. Clearly, observational studies tend to underestimate mobile phone usage while driving. The use of mobile phones and other electronic gadgets while driving has a negative effect on drivers, causing drivers to commit more driving errors. According to Redelmeier and Tribshirani [55], the risk of collision due to the use of mobile phones while driving increases four-fold compared to driving without a mobile phone. No association was established between the use of mobile phones and demographic criteria in the current study when tested with chi-square. The findings from this study were contrary to findings reported in similar studies. Most studies reported that the use of mobile phones was observed more in male drivers than in female drivers [50,56]. In order to reduce the use of mobile phones while driving in Abuja, more education and enlightenment campaigns about the hazards associated with the use of mobile phones while driving are advised to be carried out.

### 4.3.4. Use of Child Restraints

It is required that all children below the age of twelve years or not more than 135 cm in height should be restrained when in a car. Children below the ages of 7 are required to drive with a child car seat, while children between the ages of 7 and 12 are required to securely restrain themselves with a safety seat belt. The results from this study showed that 55.8% of the driving respondents admitted driving without a child car seat, with a serious risk of injuries or death if a road traffic crash was to occur. Sangowawa et al. [37], in their observational study, reported that the use of child restraints among motorists in Ibadan, Nigeria, was very low. Olufunlayo et al. [57], in an observational study, reported that just 8.5% of children in moving vehicles in Ikeja, Nigeria, were observed to be restrained. Findings from the observational studies done by other researchers have shown that the number of children restrained in moving vehicles in Abuja is fewer than the 44.2% reported in this study. The reasonably high number of respondents in this study who admitted to restraining their children while driving might be due to putting forward the appearance of socially acceptable behaviour. The high percentage of respondents who drive with children without appropriate child car seats in Abuja was observed mainly among male drivers, as the chi-square results showed that the association was statistically significant.

### 4.4. Road Safety Enlightenment Campaigns

The bivariate analysis showed that 12.3% of males and 8.3% of females indicated that they had never heard of any road safety enlightenment campaigns in Abuja. The respondents indicated they prefer receiving road safety enlightenment campaigns via radio; while 29.1% prefer radio, only 10.5% prefer daily newspapers. Although social media platforms are faster in reaching and targeting the very young population of road users, they still were less desirable, due to their potential to cause distractions while driving. Social media platforms can play a huge role in enlightenment campaigns among younger road users if they tend to read the campaign messages when they are not driving.

## 5. Conclusions

This study set out to find out the typical road user behaviours among road users in Abuja, Nigeria. The findings from this study show a high non-compliance rate for the use of seatbelts among vehicle occupants. The high non-compliance rate of seatbelt usage was mainly among young, single vehicle occupants. It was also observed that the levels of education and driving experience affect the compliance rate for the use of seatbelts. Higher levels of education and more years of driving experience significantly influence and encourage the use of seatbelts. It was also observed that a good number of drivers admitted to engaging in drink driving. Other road user behaviours such as the use of mobile phone and not using child restraints were also observed.

Based on the findings of this study, the following recommendations are suggested:

1. Agencies in charge of road safety should be better resourced to enforce road safety laws and offenders should be made to pay heavy fines if and when they violate road safety laws;
2. Equipment such as breathalyzers and speed guns should be provided for road safety agencies in order to enable them to effectively perform their duties;
3. Dissemination of road safety enlightenment campaign messages should be done over the radio, as findings from this study revealed it to be the best medium for passing information to the public.

**Author Contributions:** Conceptualization, U.N.U. and M.R.T.; methodology, U.N.U.; software, U.N.U.; validation, U.N.U.; formal analysis, U.N.U.; investigation, U.N.U.; resources, U.N.U. and M.R.T.; data curation, U.N.U. and M.R.T.; writing—original draft preparation, U.N.U.; writing—review and editing, M.R.T.; visualization, U.N.U.; supervision, M.R.T.; project administration, U.N.U. and M.R.T.; funding acquisition, U.N.U. and M.R.T. All authors have read and agreed to the published version of the manuscript.

**Funding:** This research was funded by the Petroleum Technology Development Fund (PTDF) and the School of Engineering, University of Birmingham, which sponsored the PhD tuition and maintenance fees, respectively, for Uchenna Nnabuihe Uhegbu.

**Institutional Review Board Statement:** The study was conducted according to the guidelines of the Declaration of Helsinki, and approved by the Science, Technology, Engineering and Mathematics Ethics Committee of The University of Birmingham (Application number: ERN_18_1194, approved 29 October 2018).

**Informed Consent Statement:** Informed consent was obtained from all subjects involved in the study.

**Data Availability Statement:** Not applicable.

**Acknowledgments:** The authors thank the respondents, the field assistants, and the Federal Road Safety Corps for providing the road traffic crash data used for this study.

**Conflicts of Interest:** The authors declare no conflict of interest.

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
