# Peer review of "Road User Attitudes and Their Reported Behaviours in Abuja, Nigeria"

_sustainability, doi:10.3390/su13084222_

Round 1

Reviewer 1 Report

The paper is interesting. I believe that authors aimed at addressing topical issues, and thus obtaining valuable insights for potential road safety countermeasures.

However, I see a major shortcoming - since all data were obtained from self-reporting, I would expect that authors will mention the uncertain reliability of this approach. In other words: how does self-report data reflect the observed (real-life) data? Numerous studies investigated this problem, and often found that self-reported data validity is insufficient, especially in countries with low compliance.

Example references:
Zambon et al. (2008). Seat belt use among rear passengers: Validity of self-reported versus observational measures. BMC Public Health.
Özkan et al. (2012). The validity of self-reported seatbelt use in a country where levels of use are low. Accident Analysis and Prevention.
Lipovac et al. (2015). Self-reported and observed seat belt use – A case study: Bosnia and Herzegovina. Accident Analysis and Prevention.
Lee at al. (2018). Comparing the Self-Report and Measured Smartphone Usage of College Students: A Pilot Study. Psychiatry Investigation.

Additional minor comment: some figures are not well legible.

Author Response

Reviewer 1:

The paper is interesting. I believe that authors aimed at addressing topical issues, and thus obtaining valuable insights for potential road safety countermeasures.

Point1: However, I see a major shortcoming - since all data were obtained from self-reporting, I would expect that authors will mention the uncertain reliability of this approach. In other words: how does self-report data reflect the observed (real-life) data? Numerous studies investigated this problem, and often found that self-reported data validity is insufficient, especially in countries with low compliance.

Example references:
Zambon et al. (2008). Seat belt use among rear passengers: Validity of self-reported versus observational measures. BMC Public Health.
Özkan et al. (2012). The validity of self-reported seatbelt use in a country where levels of use are low. Accident Analysis and Prevention.
Lipovac et al. (2015). Self-reported and observed seat belt use – A case study: Bosnia and Herzegovina. Accident Analysis and Prevention.
Lee at al. (2018). Comparing the Self-Report and Measured Smartphone Usage of College Students: A Pilot Study. Psychiatry Investigation.

Response 1: We have added the suggested shortcoming in the revised manuscript:

Several researchers have suggested that self – reported studies are not ideal in measuring road user behaviour due to the subjective nature and over reporting of be-haviours that would be considered not so socially acceptable [23, 26]. Naturalistic stud-ies which require that the road user behaviours be directly observed on the road are considered true representations of driver behaviours, however, self – reported methods are still used due to the high cost associated with observational studies.”

“The over – estimation of seatbelt usage among road users in self – reported studies range from a factor of 1.2 – 2, when compared to observational studies” [26, 35, 36].

References consulted include:

Helman, S. and Reed, N. (2015). Validation of the driver behaviour questionnaire using behavioural data from an instrumented vehicle and high-fidelity driving simulator. Accident Analysis and Prevention. 75: 245 – 251.

Zambon, F., Fedeli, U., Marchesan, M., Schievano, E., Ferro, A., and Spolaore, P. (2008). Seat belt use among rear passengers: validity of self – reported versus observational measures. BMC Public Health. 8 (1): 233.

Streff, F. M., and Wagenaar, A. C. (1989). Are there really shortcuts? Estimating seat belt use with self – report measures. Accident Analysis and Prevention. 21 (6): 509 – 516.

Dee, T. S. (1998). Reconsidering the effects of seat belt laws and their enforcement status. Accident Analysis and Prevention. 30 (1): 1 – 10.

Point 2: Additional minor comment: some figures are not well legible.

Response 2: All Figures and Tables are legible.

Reviewer 2 Report

The purpose of this paper is to out to investigate the road user attitudes and behaviors in Abuja, Nigeria. Using a survey campaign with a total of 1526 questionnaires (321 completed) authors tried to understand four major road user behaviors namely: use of seat belts, drink driving, use of mobile phone while driving and use of child restraints. The topic is innovative, especially in Nigeria, where the road fatality rate is very high and this type of analysis is quite interesting for policy makers. The methodology is also suitable for the research (the sample is small but representative.) Taking into account this positive assessment as a whole of the paper, I would like to mention some minor changes (or reflections) that the paper requires, mainly related to the state of the art.

  • Please, try to separate the introduction from the state of the art. In this paper the state of art is quite short and it is located inside the introduction. Try to reduce the introduction and move the state of the art to a second new chapter
  • The state of the art should include a broader state of the types of survey campaigns used to study driver user behavior and also a review of the main risk factors. There are many references to the US scenario (for example:….Ortiz et al., [11] reported that in 2015, driver distraction contributed to 3,447 deaths and 391,000 injuries in the United States. Ortiz et al., [11] in an observational study of road intersections in the United States, stated that driver distraction could be caused by single or multiple distractors). More diverse or general references should be also introduced (results from European studies, Australian, studies, Asian studies).
  • Crash fatalities in Abuja

It is very important to describe the structure and quality of the official crash data base in Nigeria. The level of quality and the breakdown of the database determines the statistical methods and the validity of the findings on road safety. The survey campaign (sample is only 321 questionnaires) may be complemented in the future with data coming from the accident data base. You can find a comparison between different national databases in the following reference. Please, introduce this reference and try to compare Nigeria data base with the rest.

Casado-Sanz, N.; Guirao, B.; Lara Galera, A.; Attard, M. Investigating the risk factors associated with the severity of the pedestrians injured on Spanish crosstown roads. Sustainability 2019, 11, 5194, https://doi.org/10.3390/su11195194

Round 2

Reviewer 1 Report

Thank you for the revision. I can see that the review comments were addressed and the paper quality has improved.

I only recommend to modify Figures 2, 3, 4 so that the text in these figures is not distorted (vertically squished).